# Effect of Culture Supernatant of *Clostridium butyricum* TO-A on Human DNA-Repair-Factor-Encoding Gene Promoters

**DOI:** 10.3390/ijms252212151

**Published:** 2024-11-12

**Authors:** Shunsuke Takaoka, Takuro Ishii, Yuriko Umihara, Ryuji Otani, Sota Akazawa, Takahiro Oda, Yoko Ogino, Yoichi Okino, Dian-Sheng Wang, Fumiaki Uchiumi

**Affiliations:** 1Department of Gene Regulation, Faculty of Pharmaceutical Sciences, Tokyo University of Science, Noda-shi 278-8510, Chiba-ken, Japan; i589toshitoshi@icloud.com (S.T.); jasudesune@gmail.com (T.I.); umhlyrk@yahoo.co.jp (Y.U.); pappywanwan37@gmail.com (R.O.); material4869@icloud.com (S.A.); 3b21028@ed.tus.ac.jp (T.O.); 3b11022@alumni.tus.ac.jp (Y.O.); 2Research Division, TOA Biopharma Co., Ltd., Tatebayashi-shi 374-0042, Gunma-ken, Japan; okino@toabio.co.jp (Y.O.); dwang@toabio.co.jp (D.-S.W.)

**Keywords:** *Clostridium butyricum*, culture supernatant, butyric acid, DNA repair, gene expression

## Abstract

In this study, *Clostridium butyricum* TO-A culture supernatant (CBCS) or butyric acid was added to a culture medium of human cervical carcinoma HeLa S3 cells, and changes in DNA-repair-related gene promoter activities were investigated. The HeLa S3 cells were transfected with a luciferase (Luc) expression vector containing approximately 500 bp of the 5′-upstream region of several human DNA-repair-related genes and cultured with a medium containing the CBCS (10%) or butyric acid (2.5 mM). The cells were harvested after 19 to 42 h of incubation. A Luc assay revealed that the human *ATM*, *PARG*, *PARP1*, and *RB1* gene promoter activities were significantly increased. A Western blot analysis showed that the amounts of the proteins encoded by these genes markedly increased. Furthermore, 8, 24, and 48 h after the addition of the CBCS (10%), total RNA was extracted and subjected to RNAseq analysis. The results showed that the expression of several inflammation- and DNA-replication/repair-related genes, including *NFKB* and the *MCM* gene groups, decreased markedly after 8 h. However, the expression of the histone genes increased after 24 h. Elucidation of the mechanism by which the CBCS and butyrate affect the expression of genes that encode DNA-repair-associated proteins may contribute to the prevention of carcinogenesis, the risk of which rises in accordance with aging.

## 1. Introduction

It has been suggested that defects in DNA repair control, which is thought to be associated with cellular senescence, affect carcinogenic mechanisms [1]. We established an experimental system to evaluate the promoter activities of the 5′-upstream regions containing transcription start sites (TSSs) of well-known human genes whose encoding proteins are involved in cell growth suppression and DNA repair functions [2,3]. We have reported that the natural compound *trans*-resveratrol (Rsv) markedly increased *TP53* [4], *HELB* [5], and *MCM4* [6] gene promoter activities in HeLa S3 cells. Finding beneficial natural compounds that are contained in food will contribute to the development of novel therapies for treating cancer with fewer side effects.

The gut microbiota not only play an important role in food digestion and nutrient metabolism but also have beneficial effects on human health, including the immune [7] and nervous systems [8]. In this study, we first examined whether culture supernatant of probiotic bacteria (*Bacillus subtilis* TO-A (BS), *Enterococcus faecium* T-110 (EF), and *Clostridium butyricum* TO-A (CB)) increased the promoter activities of genes encoding proteins that are involved in cell growth suppression and DNA repair systems. It is considered important for maintaining health that these three types of bacteria used all grow in a well-balanced manner in the human intestine [9]. For example, *B. subtilis* TO-A could extend the lifespan of *C. elegans* [10]. Certain strains of *E. faecium* were identified to have favorable effects on immunomodulatory properties [11]. The beneficial health effect of *C. butyricum* has also been investigated [12]. This study revealed that the culture supernatant of *C. butyricum* TO-A (CBCS) significantly enhanced the promoter activities of several well-known DNA-repair-related genes in HeLa S3 cells. *C. butyricum* is contained in fermented foods and produces n-butyric acid, which is found in various foods, such as fermented butter. It has been suggested that these bacteria have beneficial effects for maintaining good health conditions. In addition, n-butyric acid is known to suppress cancer cell growth and inflammation, inducing changes in cell morphology [13], promoting cell death, and triggering apoptosis in a colon cell line [14]. Butyrate induces acetylation not only of histones but also of the GC-box-recognizing transcription factor Sp1, affecting specific gene expression [14]. In this study, we examined whether the addition of n-butyric acid (5 mM) to a HeLa S3 medium instead of the CBCS would activate certain DNA-repair-related gene promoters.

Techniques for analyzing intracellular transcripts have already been developed, and in this study, HeLa S3 cells were cultured in a medium containing the CBCS (10%), and an RNAseq analysis was carried out. The results showed that accumulation of the transcripts did not always correlate with promoter activation. The expression of some of the DNA-replication-factor-encoding genes had significantly decreased by 8 h after the CBCS treatment. The expression of certain specific DNA-repair-factor-encoding genes whose promoters have “duplicated GGAA sequences” [15] could be activated in response to CBCS- or butyrate-induced signals. Activation of the promoters might be required to overcome a decrease in the transcripts. Our results suggest that post-transcriptional regulation occurred for each gene’s expression, even when promoter activity was induced by butyrate. Elucidation of the post-transcriptional regulation mechanism by the CBCS or butyrate will enable clinical applications to prevent aging-associated carcinogenesis.

## 2. Results

### 2.1. Screening of Probiotic Bacterial Culture Supernatant That Affects Promoter Activities of Human DNA-Repair-Factor-Encoding Genes

Recent studies have revealed that intestinal bacteria or metabolites affect various health conditions [7,8]. We studied chemically synthesized and natural compounds, including *trans*-resveratrol, that up-regulate human genes encoding DNA repair and replication factors [3,4,5,6], and based on these observations, we hypothesized that some metabolites of microorganisms beneficial to health might affect the transcription of genes encoding DNA-repair-associated factors. Previously, we developed and established a Luc reporter assay that could evaluate the promoter activities of multiple human genes, including DNA-repair- and mitochondrial-function-associated genes [2,3]. DNA-transfected cells were harvested between 19 and 42 h after the addition of a culture medium containing 10% BS, EF, CB or *B. amyloliquefaciens* TOA5001 (BA) culture supernatant (CS), and the cellular extracts were prepared for the Luc assay (Figure 1A and Appendix A). The results indicated that the CBCS had the most apparent effect on inducing the promoter activities of several human DNA-repair-factor-encoding genes, including the *ATM*, *PARG*, *PARP1*, and *RB1* genes. The other bacteria-cultivated supernatants had no such effect on inducing promoter activities. Thereafter, we focused on analyzing the effect of the CBCS on the DNA-repair-factor-encoding gene promoters. The induction of the promoter activities by the 10% CBCS-containing medium was greater at 42 h than at 19 h after the treatment of the cells (Figure 1B).

### 2.2. Morphological Changes in and Survival of HeLa S3 Cells After the Addition of Probiotic-Bacteria-Cultivated Supernatants

Morphological changes after the addition of 10% CBCS or a CBCS/BSCS/EFCS mixture (CSM) were confirmed. After 19 h of incubation, the cell shapes had changed to be longer than those in the control (Figure 2A). Changes in both the shape and relative number of cells were apparent after further incubation with the CBCS (Figure 2B). The relative cell number after 48 h of cultivation with 10% CBCS was reduced to less than 20% of that with the control bacteria-culture-containing medium (Figure 3). The growth of the HeLa S3 cells was greatly reduced by cultivation with a higher concentration of the bacterial culture medium. Thereafter, the concentration of the CBCS or the control bacterial culture medium was set as 10%. The morphological changes were also confirmed after n-butyric acid treatment (Figure 2C). The observations were consistent with a previous report showing that the addition of n-butyric acid to a culture medium caused morphological alterations in HeLa cells [13].

### 2.3. Responses of Human DNA-Repair-Factor-Encoding Gene Promoters to n-Butyric Acid

Both the induction of morphological changes in the HeLa S3 cells and a suppressive effect on their proliferation could similarly be observed after the addition of the n-butyric acid to the culture medium. We therefore speculated whether the n-butyric acid, which was the unique metabolite in the CB-cultivated broth, might affect the promoter activities of DNA-repair-factor-encoding genes. The treatment with n-butyric acid (2.5 mM) had induced several gene promoter activities at 19 h (Figure 4). The magnitude of the induction of the *ATM*, *RB1*, and *PARP1* promoters’ activities by the addition of n-butyric acid (5 mM) after 42 h was the most prominent of all (Appendix A). It should be noted that the *TP53* promoter was not affected as much by the n-butyric acid treatment. The fold activation of the *TP53* promoter’s activity by 1.25, 2.5, and 5 mM after 19 h was 0.298, 0.819, and 1.59, respectively, suggesting that the n-butyric acid could induce a regulatory signal(s) that suppressed the *TP53* promoter’s activity.

### 2.4. Amounts of ATM, PARP1, and RB1 in the HeLa S3 Cells After CBCS Treatment

The amounts of ATM, PARP1, and RB1, for which the encoding gene promoter activities were greatly induced by the addition of the CBCS (10%), were analyzed by Western blotting (Figure 5A). The ATM (>200 kDa) protein was detected only at a low level in the control cell extract, but it was induced by 2 h after addition of the CBCS (10%) and peaked at 24 h (Figure 5A, top). Similarly, the RB1 (106-kDa) protein was induced and peaked at 8 to 16 h (Figure 5A, second). The two bands that represent 120 and 105 kDa PARP1 proteins were hardly detected in the control cell extract. Both bands were induced by 1 h and had mostly accumulated by 8 h after the addition of the CBCS (10%) (Figure 5A, bottom). Although β-actin (ACTB, 50 kDa) was apparently detected in the 0 h control cell extract, it was gradually up- and down-regulated from 1 to 48 h after the addition of the CBCS (10%) (Figure 5A, third). A similar experiment was carried out after the addition of the n-butyric acid (2.5 mM) to the HeLa S3 cell culture medium (Figure 5B). Basically, the induction of the ATM, RB1, and PARP1 proteins was observed to be similar to that after the addition of 10% CBCS. However, they accumulated earlier than they did under the CBCS (10%) treatment. The amounts of ATM, RB1, and PARP1 peaked at 2, 4, and 2 h after the addition of n-butyric acid (Figure 5B).

### 2.5. n-Butyric Acid Response Elements in the Human RB1 and PARP1 Gene Promoters

Because the *RB1* and *PARP1* gene promoters most prominently responded to the CBCS (10%) (Figure 1B and Appendix A) and n-butyric acid (Appendix A) among the human gene promoters tested, deletion experiments were carried out (Figure 6). Deletion identified constructs pGL4-RB1Δ3 and pGL4-PARP1Δ2 as those that responded to n-butyric acid (5 mM) most positively (Figure 6A). Both 76 and 90 bp sequences which are contained in the pGL4-RB1Δ3 and pGL4-PARP1Δ2 plasmids, respectively, have a duplicated GGAA (TTCC) motif. Notably, the JASPAR-2020 analysis indicated that the SPI1 (PU.1) recognition sequence was contained in both the 76 and 90 bp sequences that responded to butyrate (Figure 6B).

### 2.6. RNA Sequence (RNAseq) Analysis of the HeLa S3 Cells After Cultivation with the CBCS

To evaluate the genes transcriptionally activated and inactivated by the addition of the CBCS (10%) to the HeLa S3 culture medium, an RNAseq analysis was performed (Figure 7). In total, 6669, 6727, and 6641 genes were identified as differentially expressed between the CBCS group and the control group after 8, 24, and 48 h of cultivation (Appendix A). Among these genes, 4248, 4397, and 4719 genes were significantly up-regulated, and 2421, 2330, and 1922 genes were significantly down-regulated 8, 24, and 48 h after the addition of CBCS (10%), respectively. A number of these genes were further classified as protein-coding or non-coding genes (Figure 8). In summary, the results showed that significantly up-regulated genes were dominant after the addition of CBCS (10%) (Figure 7 and Appendix A). The ratio of non-coding/protein-coding genes was 50% at 8 h, and then it declined to 30% after 24 h (Figure 8).

Gene Ontology (GO) annotation classified the DEGs compared with the control at the same time point (Figure 9). At 8 h after the addition of the CBCS (10%), the most significant GO terms were transmembrane/synaptic signaling and transcription factor activity (Figure 9A), for which most of the genes were up-regulated (Appendix A). However, the down-regulated genes were mainly correlated with transcription factor activity and transcriptional activator activity. The differentially expressed genes (DEGs) that were up-regulated after 24 and 48 h of treatment with the CBCS (10%) were different from those up-regulated in the cells treated for 8 h. The significant GO terms after 24 and 48 h of CBCS (10%) treatment were extracellular matrix/structure organization and various ion-channel-related genes (Figure 9B,C). Most of the genes that were classified as having a significant GO term were up-regulated (Appendix A). According to the KEGG analyses between the 8, 24, and 48 h experiments, the DEGs that were commonly listed as significant are included in pathways related to neuroactive ligand–receptor interactions and cytokine–cytokine receptor interactions (Figure 10). We additionally compared expressions of the DNA-repair-associated genes from the original data set (Table 1).

## 3. Discussion

In this study, it was shown that several human DNA-repair-associated gene promoters prominently responded to the addition of the CBCS to the human cervical carcinoma HeLa S3 cells. In contrast, the addition of the other probiotic bacterial culture supernatants, including the BSCS, EFCS, and BACS, to the HeLa S3 cell culture did not cause up-regulation at all (Figure 1A). Interestingly, the promoter activities of most of the genes were decreased in the BACS (10%)-treated cells (Figure 1A, purple bars). Therefore, the expression of the DNA-repair-factor-encoding genes could be suppressed simultaneously by the BACS treatment. Although activities of the *RB1* and *TP53* promoters were detected, those of the *PARP1* and *TERT* promoters were almost zero, implying that the compounds in the BACS may be applied for developing cell-death-inducing cancer therapeutics.

The enhancement in the promoter activities increased in accordance with the morphological changes in the cells (Figure 1B and Figure 2A,B). In this experimental setting, morphological changes in the HeLa S3 cells [13] were observed after the addition of n-butyric acid to the culture medium (Figure 2C), and some of the DNA-repair-factor-encoding gene promoters apparently responded (Figure 4). In addition, the CBCS and n-butyric acid were able to cause the ATM, RB1, and PARP1 proteins to accumulate in the HeLa S3 cells (Figure 5).

We have studied promoters of DNA-repair-factor-encoding genes including *HELB* [5], *MCM4* [6], and *CDC45* [16]. Notably, they commonly contain GGAA (TTCC) motifs and GC-boxes, which are ETS family [17] and Sp1 [18] recognition sequences, respectively. All of them responded to a natural compound, *trans*-resveratrol (Rsv), in HeLa S3 cells [5,6,16]. In this study, deletion experimentation showed that the n-butyric acid-responding regions in the human *RB1* and *PARP1* promoters contain duplicated GGAA (TTCC) motifs (Figure 6). Although the JASPAR-2020 analysis did not predict the GC-boxes in these n-butyric acid-responding regions, the GC-rich sequences 5′-CGGGCGGGGG-3′ and 5′-CGGCGGTGGCCGG-3′ were present in the *RB1* and *PARP1* promoters, respectively (Figure 6B). The implication that Sp1 could act as a transcription activator is consistent with a previous study that indicated that Sp1-binding elements in the human *ANT2* promoter are required for HeLa cells to respond to butyrate [19]. Moreover, three Sp1-binding sequences that are contained in the HIV LTR are required for a positive response to butyrate in HeLa cells [20]. The possibility that the GC-box-like sequences and GGAA duplication cooperatively confer the positive response to n-butyric acid is yet to be elucidated.

Butyrate is a histone-deacetylase (HDAC) inhibitor [21] that impairs pancreatic β-cell function [22,23]. The butyrate, a short-chain fatty acid could drastically increase the level of histone acetylation [24] in HeLa cells to inhibit chromosomal condensation [25], up-regulating specific gene expression [26,27]. Moreover, butyrate is produced by bacteria in the gut [28]. It not only suppresses tumor genesis but also viral DNA replication [29,30,31]. Additionally, the CBCS prominently reduced the expression of the *NFKB* gene, which can suppress inflammation (Table 2). The RNAseq data from this study indicated some novel findings. Firstly, the up-regulated gene expression was two-fold that of down-regulated genes at all time points tested (Appendix A). Secondly, the number of non-coding RNAs (ncRNAs) was most abundant 8 h after the addition of the CBCS (Figure 8), suggesting a specific transcription system worked to evoke the non-coding RNA expression at an early stage of the response to the CBCS or the n-butyric acid. It must have lowered the relative expressions of the protein-encoding-genes. Although some specific ncRNAs may affect the translation efficiency, they are not directly used as templates to produce polypeptides. Therefore, each level of protein would simply be dependent on the expression of its coding gene. It is consistent that the levels of the ATM, RB1, and PARP1 proteins were increased after the n-butyric acid treatment. However, the raw RNAseq data showed that most of the genes for which the promoter activities were up-regulated by the CBCS or n-butyric acid (Figure 1 and Figure 4) were not increased as expected (Table 1). When all the data on the non-coding RNAs were eliminated, some were identifiable as significantly induced. We used the *PIF1* promoter as a control because its activity was relatively high, but it showed very low responses to the CBCS and the n-butyric acid among the promoters tested. Surprisingly, its expression level (FPKM value) in the CBCS-treated cells was significantly decreased compared with that in the control cells (Table 1). When it was used as a reference control, most of the relative expressions of the genes in Table 1, except for that of *TP53*, were identifiable as increased. Both the *TP53* and *NFKB1* transcripts were evaluated as significantly decreased after 8 to 48 h (Table 2). p53 is generally known as a tumor suppressor that inhibits cell proliferation to induce apoptosis or cell death. Although HeLa S3 cells have wild-type *TP53* in their chromosomes [32], quite a lot of cancers have mutations in the protein-encoding sequences of the gene [33]. In this regard, down-regulation of the gene expression by the CBCS could be applied to *TP53*-mutated cancer, suppressing the accumulation of mutated p53. Our results also support previous studies showing that butyrate inhibits NF-κB activation in human epithelial cells [34] and in degenerated intervertebral disc tissues [35]. Also, in mice, butyrate can inhibit NF-κB activation [36,37], suggesting that it could limit immune signals, including inflammatory response. Finally, although it was not indicated by either the GO term or KEGG analyses, the expression of well-known protein-encoding genes was identified. Although molecular mechanisms that would explain how histone-encoding genes are transcribed are not yet fully understood, many of them are head–head-linked with oppositely transcribed histone-encoding genes [38]. Several histone-encoding gene transcripts were up-regulated after 8 h of the CBCS treatment (Table 2). This result is consistent with a previous report indicating that butyrate induces poly(A)-H1 histone mRNA in HeLa cells [39]. Additionally, histones H2B, H3, and H4 [40] are hyperacetylated in a transcriptionally active mononucleosome-rich fraction of HeLa S3 cell nuclei [41]. Moreover, hypermethylation was caused in the active nucleosomal DNA [42]. Therefore, the reason why specific genes, including histone-encoding genes, are activated at an early stage after butyrate treatment could partly be explained by alterations in the local structures of the chromosomes. In contrast, DNA-replication-factor-encoding gene transcripts, including *NFKB1*, *PCNA*, *POLA1*, *POLE*, *POLF*, and *MCM2/3/5/7*, were significantly down-regulated (Table 2). This observation is consistent with a previous report showing that butyrate suppresses DNA replication to enhance etoposide-induced human tumor cell death [43]. In addition, the up-regulated RNAs after 24 and 48 h were transcripts from genes that encode immune-response-associated factors. The mechanisms that induced the expression of genes should be elucidated.

## 4. Materials and Methods

### 4.1. Materials

The n-butyric acid (butyrate) (Cat. No. 023-05396) was purchased from Fujifilm (Tokyo, Japan).

### 4.2. Preparation of the Probiotic Bacterial Culture Supernatants

For the probiotic bacterial culture, appropriate nutrient media containing peptone, sugar, etc., were used as the manufacturing media to culture each strain listed below. The nutrient medium was used as a negative control in each experimental setting in this study, and the probiotic bacterial culture supernatants were prepared from each strain-cultured suspension. In short, *B. subtilis* TO-A (BS), *E. faecium* T-110 (EF), *C. butyricum* TO-A (CB), and *Bacillus amyloliquefaciens* TOA5001 (BA) from Toa Biopharma Co., Ltd. (Tokyo, Japan) were cultured with each nutrient medium at 37 °C for the appropriate culture times, respectively. Only CB was cultured under anaerobic conditions. Then, the cultured suspensions were filtrated through hydrophilic PVDF membrane filters with a diameter of 47 mm and a pore size of 0.22 µm (Durapore^®^, Merck Millipore Ltd., Cork, Ireland).

### 4.3. Cells and Cell Culture

Human cervical carcinoma (HeLa S3) cells [2] were grown in Dulbecco’s Modified Eagle’s Medium (DMEM Low Glucose) (Nakarai Tesuque, Kyoto, Japan) supplemented with 10% fetal bovine serum (FBS) (Biosera, East Sussex, UK) and penicillin–streptomycin at 37 °C in a humidified atmosphere with 5% CO_2_.

### 4.4. Construction of THE Luciferase (Luc) Reporter Plasmids

The Luc reporter plasmids, carrying approximately 500 bp, that contained transcription start sites (TSSs) of human DNA-repair- and cell-cycle-control-factor-encoding genes were constructed previously. The promoter regions that are contained in the plasmids, pGL4-PARG D6 [44], pGL4-HDHB [5], and pGL4-BLM [45], were characterized previously. The pGL4-ATM, pGL4-ATR, pGL4-p21, pGL4-RB1, pGL4-WRN, pGL4-BRCA1, pGL4-PARP1, pGL4-E2F4, pGL4-TP53, and pGL4-TERT plasmids have been shown to contain functional promoters [3,46,47]. They were used previously for multiple DNA transfection using the DEAE-dextran method [2]. Deletion of the *RB1* and *PARP1* promoters was carried out via PCR using sense and anti-sense primers (Table 3) and pGL4-RB1 [46] and pGL4-PARP1 [47], respectively, as a template. The amplified DNA fragments were digested with *Kpn*I and *Xho*I, and they were ligated into the MCS of pGL4.10[*luc*2] to generate pGL4- RB1-Δ1, pGL4-RB1-Δ2, pGL4-RB1-Δ3, pGL4-PARP1-Δ1, and pGL4-PARP1-Δ2. The nucleotide sequences were confirmed by a DNA sequencing service (FASMAC, Greiner Japan Inc., Atsugi, Japan) with the primers Rv (TAGCAAAATAGGCTGTCCCC) and GL (CTTTATGTTTTTGGCGTCTTCC). The Luc reporter plasmid, pGL4-PIF1, was constructed as described in [47].

### 4.5. Transient Transfection and the Luciferase (Luc) Assay

The Luc reporter plasmids were transfected into the HeLa S3 cells in 96-well plates using the DEAE-dextran method [2], and after 24 h of transfection, the culture medium was changed to DMEM with 10% FBS containing 10% probiotic bacterial culture supernatant or the control culture. After a further 24 h of incubation, the cells were collected and lysed with 100 μL of 1 × cell culture lysis reagent, containing 25 mM Tris-phosphate (pH 7.8), 2 mM DTT, 2 mM 1,2-diaminocyclohexane-N,N,N′,N′,-tetraacetic acid, 10% glycerol, and 1% Triton X-100, and then mixed and centrifuged at 12,000× *g* for 5 s. The supernatant was stored at −80 °C. The Luc assay was performed using a Luciferase assay system (Promega, Madison, WI, USA), and the relative Luc activities were calculated as described previously [5,15,44,45,46,47].

### 4.6. Western Blot Analysis

A Western blot analysis was carried out after SDS-PAGE (15% acrylamide) as previously described [4,5,6], using antibodies against PARP1 (Cat. No. sc-7150), RB1 (Cat. No. sc-50, Santa Cruz Biotechnology, Santa Cruz, CA, USA), and β-actin (Cat. No. A5441, Sigma-Aldrich, St Louis, MO, USA), followed by the addition of horseradish peroxidase (HRP)-conjugated anti rabbit (Cat. No. A0545) or anti mouse IgG (Cat. No. A9917) secondary antibodies (Sigma-Aldrich). The signal intensities were quantified using the ChemiDoc and ImageLab systems (BioRad, Berkeley, CA, USA).

### 4.7. The Cell Viability Assay (the CCK-8 Assay)

A cell viability assay was carried out using a CCK-8 assay kit (TOYOBO, Tokyo, Japan), according to the manufacturer’s protocol. Briefly, after 48 h of cultivation of the HeLa S3 cells with the CBCS or a bacterial culture medium containing DMEM/10% FBS, CCK-8 (10 μL) was added to each well. After a further 1 h of incubation, A_450_ was measured using a SYNERGY HTX (Agilent Technologies, Santa Clare, CA, USA).

### 4.8. Statistical Analysis

The standard deviation (S.D.) for each data point was calculated, and the results are shown as the means ± S.D. from three independent experiments. Statistical analysis of the data shown in Figure 3 was performed using Student’s *t*-test, and asterisks indicate values for which ** *p* < 0.01.

### 4.9. Preparation of Total RNA from the HeLa S3 Cells and RNAseq Analysis

After 0, 8, 24, and 48 h of cultivation with DMEM/10% FBS containing 10% CBCS or the control bacterial culture medium, the HeLa S3 cells were harvested, RNAs were isolated using the GenElute^TM^ Mammalian Total RNA Miniprep kit (SIGMA Aldrich Japan Co. Ltd., Tokyo Japan), and the quality was analyzed using an Agilent 2100 bioanalyzer (Agilent technologies, Santa Clare, CA, USA). The RNA sequencing and data analyses were carried out by Novogene Co. Ltd. (Singapore). The gene expression profiles of the CBCS and control groups were analyzed using the Illumina HiSeq2000 RNA Sequencing system (Illumina, San Diego, CA, USA). Original image data files from the high-throughput sequencing platforms (Illumina) were transformed into sequenced reads (raw data or raw reads) using CASAVA base recognition (base calling). The raw data were stored in FASTQ(fq) format files which contained sequences of the reads and the corresponding base quality. They were filtered to remove low-quality reads or reads with adaptors. FPKM (fragments per kilobase of transcript sequence per million base pairs sequenced) values were applied to estimating the gene expression levels, which took the effects of both sequencing depth and gene length on the counting of the fragments into consideration [48].

### 4.10. Identification of Differentially Expressed Genes (DEGs)

Differential gene expression analysis was carried out by Novogene. Briefly, after quantification of the gene expression, a statistical analysis of the expression data was carried out to indicate the differentially expressed genes (DEGs) and significantly expressed genes between two groups where |log2(FoldChange)| > 1 and padj < 0.05.

### 4.11. Functional Enrichment Analysis

The clusterProfiler 3.8.1 [49] software was applied to perform an enrichment analysis, including GO, KEGG, and Reactome [50] database enrichment.

## 5. Conclusions

The probiotic *C. butyricum* has been revealed to have beneficial health effects, including anti-cancer activity [12,51,52]. Although we used HeLa S3 cells in this study, butyrate has effects that serve to prevent tumor genesis, viral replication, and inflammation in the gut [28]. However, we should not ignore the fact that the microbiota, including CB, produce not only butyrate but also other compounds beneficial for health [53].

ncRNAs [54], including micro RNAs (miRNAs), which are contained quite abundantly in cells, have relevant biological functions, even affecting cell behavior [55]. Moreover, long ncRNAs (lncRNAs), including *MALAT1*, *NEAT1*, and *NORAD*, have been revealed to have biologically important functions [56]. Notably, some specific miRNAs might be associated with tumor suppression [57,58]. In this study, many ncRNAs were up-/down-regulated by the CBCS treatment. Some of them may play key roles in evoking morphology-changing signals to arrest the cell cycle. Finding the essential ncRNAs that stop cell proliferation but maintain integrity in the DNA repair system will contribute to developing novel drugs for cancer therapy with lower side effects.

## Figures and Tables

**Figure 1 ijms-25-12151-f001:**
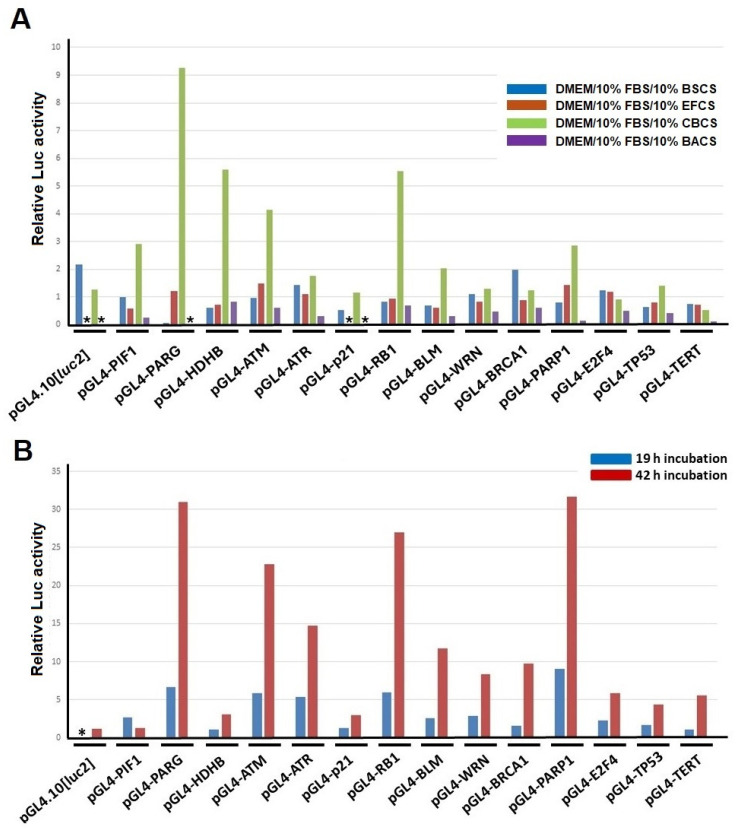
Responses of human gene promoters to bacteria-cultivated supernatants in HeLa S3 cells. (**A**) Screening of bacteria-cultivated supernatants (BCSs) that could activate human DNA-repair-factor-encoding gene promoters. BSCS: *Bacillus subtilis* TO-A culture supernatant; EFCS: *Enterococcus faecium* T-110 culture supernatant; CBCS: *Clostridium butyricum* TO-A culture supernatant; BACS: *Bacillus amyloliquefaciens* TOA5001 culture supernatant. (**B**) Effect of CBCS on human DNA-repair-factor-encoding gene promoters. Three independent experiments were carried out. Results show relative Luc activities compared with those of the pGL4-PIF1-transfected cells untreated with BCS. Asterisks indicate values that were not determined.

**Figure 2 ijms-25-12151-f002:**
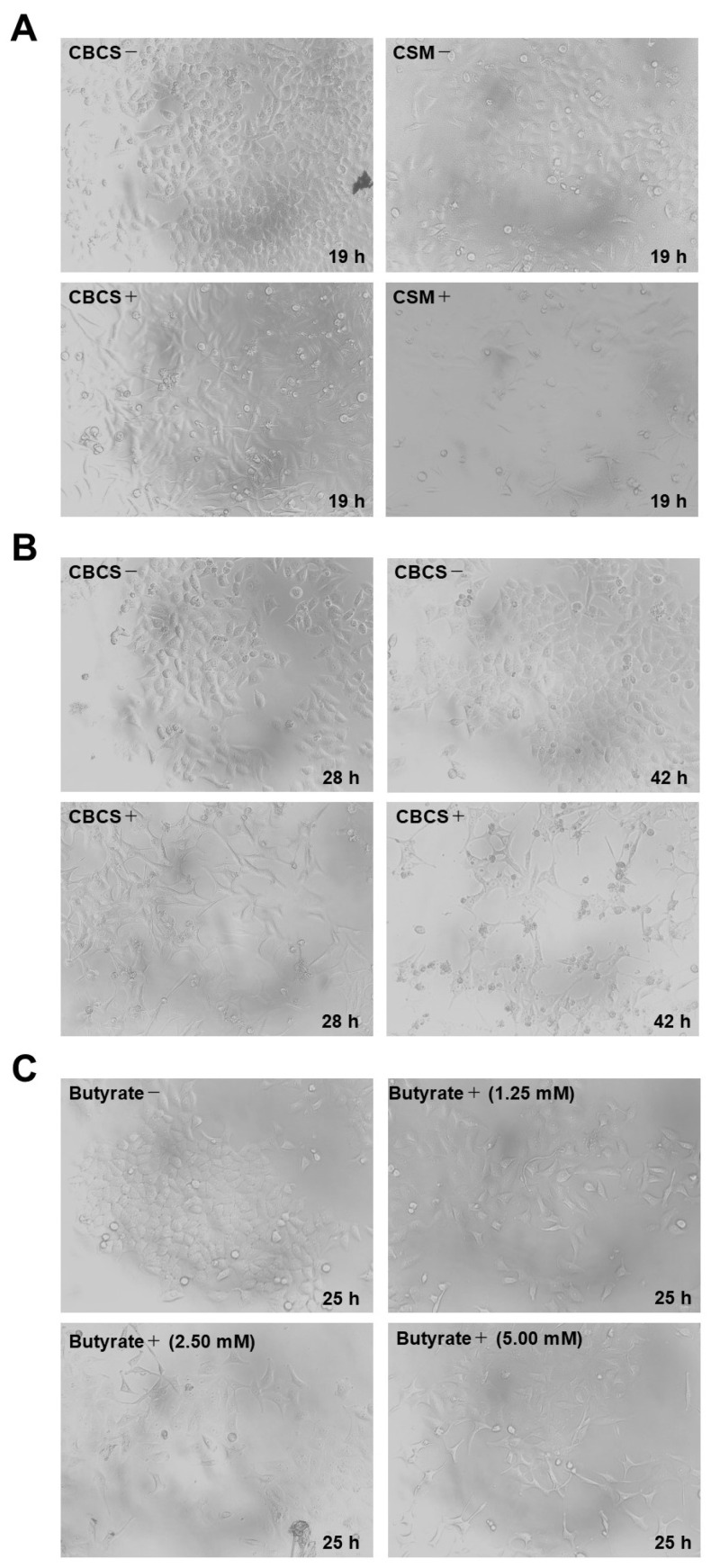
Morphological changes in the HeLa S3 cells after treatment with BCSs. (**A**) HeLa S3 cells were cultured in a medium containing 10% CBCS or CBCS/BSCS/EFCS mixture (CSM) for 19 h (lower panels). The upper panels indicate HeLa S3 cells that were cultured in a medium with 10% control supernatant for 19 h. (**B**) HeLa S3 cells were cultured in a medium containing 10% CBCS for 28 and 42 h (lower panels). The upper panels indicate HeLa S3 cells that were cultured in a medium with 10% control supernatant for 28 and 42 h. (**C**) HeLa S3 cells were cultured in a medium containing 0 (upper left), 1.25 (upper right), 2.5 (lower left), and 5 mM (lower right) of n-butyric acid for 25 h.

**Figure 3 ijms-25-12151-f003:**
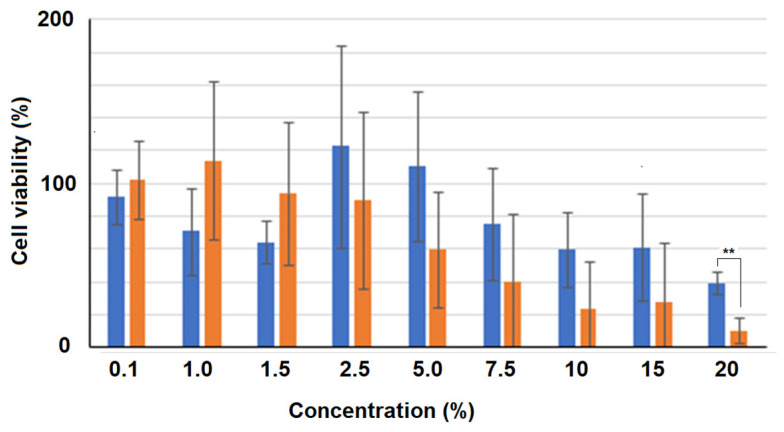
The inhibitory effect of the CBCS on HeLa S3 cell proliferation. HeLa S3 cells (2500 cells/well) were cultivated in a 96-well plate for 24 h. Then, the culture medium was changed to that containing 0 to 20% of the CBCS (orange columns) or a control bacterial culture medium (blue columns), and a CCK-8 assay was carried out after 48 h of incubation at 37 °C with 5% of CO_2_. The results are shown as means ± SD from three independent experiments. Statistical analysis was performed with Student’s *t*-test, and asterisks (**) indicate a value of ** *p* < 0.01.

**Figure 4 ijms-25-12151-f004:**
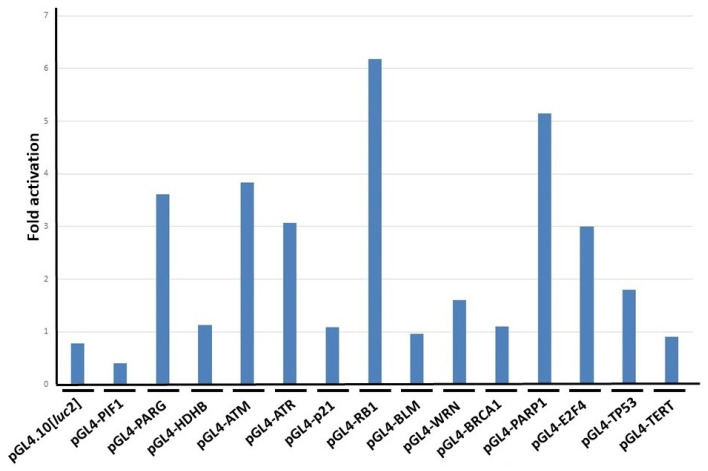
Responses of the human gene promoters to n-butyric acid in HeLa S3 cells. HeLa S3 cells were transfected with Luc reporter plasmids, including pGL4.10[*luc*2] and pGL4-PIF1 as negative and positive control vectors, respectively. After 4 h of transfection, the culture medium was changed to that containing 2.5 mM of n-butyric acid. After further incubation, the cells were corrected, and the Luc assay was carried out. Averages from three independent experiments with or without n-butyric acid were calculated. Fold activation indicates the ratio of the results for the average Luc activity of the butyrate-containing culture medium to those for the culture medium that did not contain butyrate.

**Figure 5 ijms-25-12151-f005:**
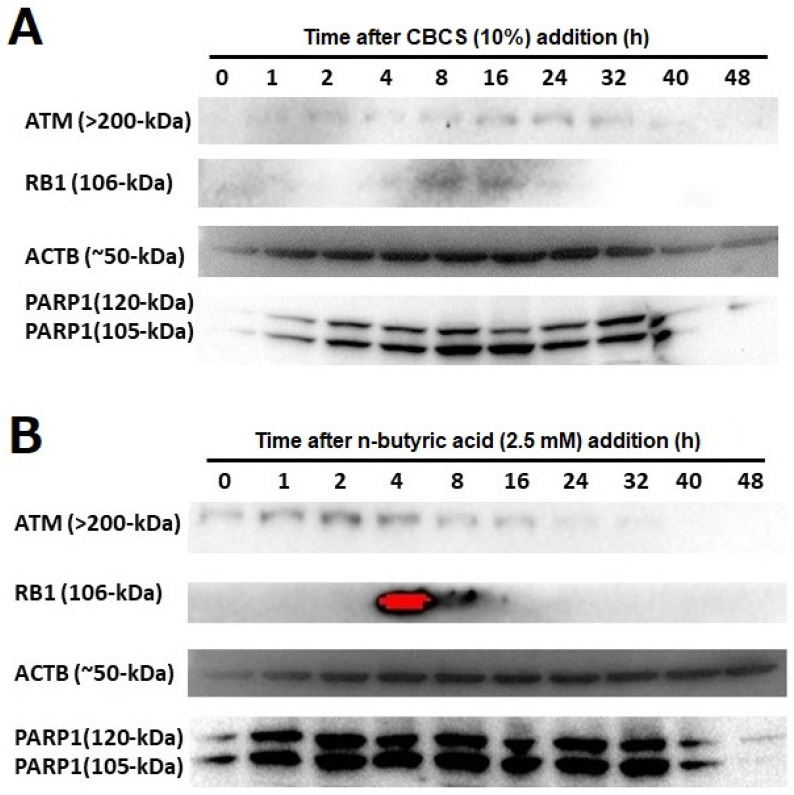
Levels of ATM, RB1, and PARP1 proteins in HeLa S3 cells. The HeLa S3 cells (1 × 10^6^) were cultivated with a DMEM containing 10% FBS for 24 h. Then, the culture medium was changed to that containing (**A**) 10% CBCS or (**B**) n-butyric acid (2.5 mM). Zero to forty-eight hours after the medium exchange, the cells were corrected, and RIPA buffer extracts were subjected to SDS-PAGE and Western blotting.

**Figure 6 ijms-25-12151-f006:**
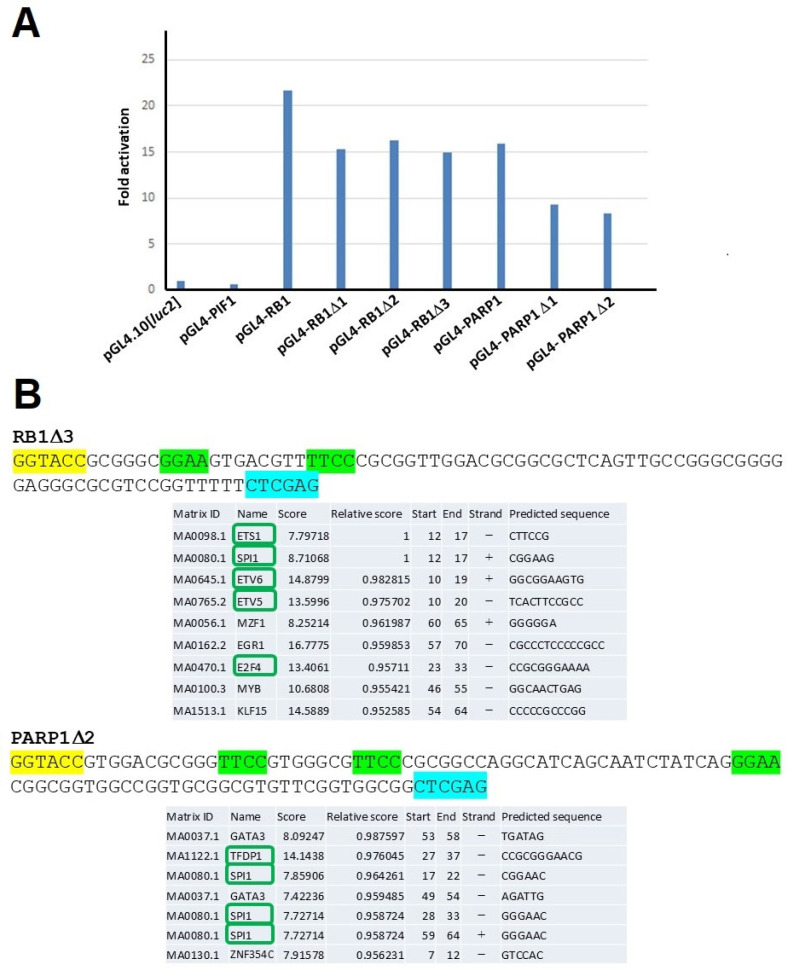
n-Butyric acid response elements in the human *RB1* and *PARP1* gene promoters. (**A**) Deletion experiments on the human *RB1* and *PARP1* promoters. HeLa S3 cells were transfected with Luc reporter plasmids. After 4 h of transfection, the culture medium was changed to that containing 5 mM of n-butyric acid, and similar experiments were carried out. Averages from three independent experiments with or without n-butyric acid were calculated. Fold activation indicates the ratio of the results for the average Luc activity of the butyrate-containing culture medium to those for the culture medium that did not contain butyrate. (**B**) n-Butyric acid-responsive core sequences in the 5′-upstream regions of human *RB1* and *PARP1*. The n-butyric acid-responsive sequences in pGL4-RB1Δ3 and pGL4-PARP1Δ 2 were applied in the JASPAR-2020 program (with a threshold > 95%). Restriction sites for *Kpn*I and *Xho*I enzymes are highlighted in yellow and pale blue, respectively. The green-highlighted GGAA and TTCC are the core motifs that are recognized by transcription factors, including ETS family proteins.

**Figure 7 ijms-25-12151-f007:**
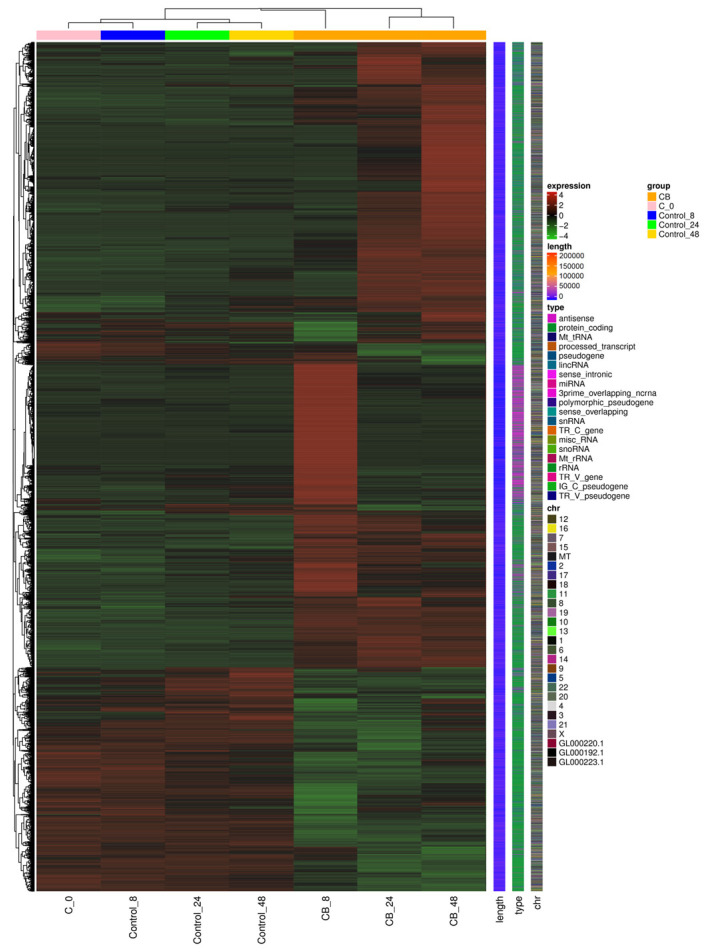
Heat map for the RNAseq cluster analysis of differentially expressed genes (DEGs) between samples. HeLa S3 cells that were cultivated with or without CBCS (10%) for 0, 8, 24, and 48 h. Red and green represent up- and down-regulated genes, respectively.

**Figure 8 ijms-25-12151-f008:**
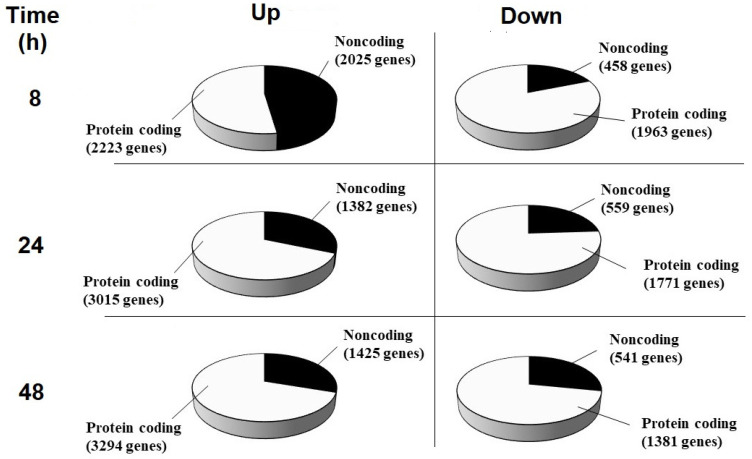
Classification of genes by protein coding. The up- (**left**) and down-regulated (**right**) genes of HeLa S3 cells cultivated for 8 (**upper**), 24 (**middle**), and 48 h (**lower**) were classified further as protein-coding (white pie portions) or non-coding RNAs (black pie portions).

**Figure 9 ijms-25-12151-f009:**
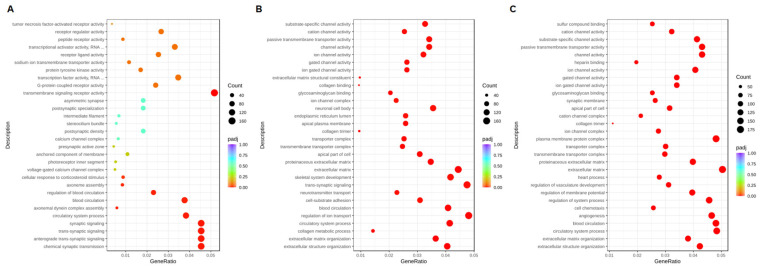
GO enrichment analysis comparing differential genes between CBCS-treated and non-treated HeLa S3 cells. RNA samples were obtained after (**A**) 8, (**B**) 24, and (**C**) 48 h of cultivation with or without CBCS (10%).

**Figure 10 ijms-25-12151-f010:**
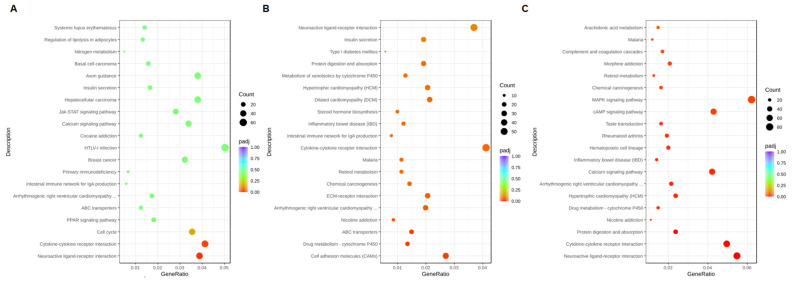
KEGG enrichment analysis comparing differential genes between CBCS-treated and non-treated HeLa S3 cells. RNA samples were obtained after (**A**) 8, (**B**) 24, and (**C**) 48 h of cultivation with or without CBCS (10%).

**Table 1 ijms-25-12151-t001:** Comparisons of FPKM values between CBCS-treated and untreated HeLa S3 cells.

Time (h)		8			24			48		
Gene	CBCS	FPKM	Fold	*p*-Value	FPKM	Fold	*p*-Value	FPKM	Fold	*p*-Value
*PIF1*	−	4.243928	0.896654	0.604697	5.52694	0.123736	4.59 × 10^−11^	4.295748	0.137542	5.07 × 10^−10^
	+	3.805336			0.683883			0.590845		
*PARG*	−	8.055117	0.494794	0.013937	6.682184	1.014106	0.902802	7.401787	0.886953	0.547691
	+	3.985623			6.776441			6.565034		
*HELB*	−	1.034974	0.921202	0.748394	0.934543	0.471383	0.017126	0.861079	0.743293	0.290985
	+	0.95342			0.440527			0.640034		
*ATM*	−	3.14527	0.736635	0.218432	3.311033	1.050696	0.998052	3.869949	0.891405	0.553431
	+	2.316915			3.47889			3.449691		
*ATR*	−	6.495218	0.579147	0.040087	5.301323	0.712684	0.176728	5.465636	0.687106	0.13576
	+	3.761683			3.77817			3.75547		
*CDKN2A*	−	40.86149	0.66904	0.113326	44.3764	0.30763	2.88 × 10^−5^	35.56736	0.781209	0.291049
	+	27.33799			13.65149			27.78555		
*RB1*	−	18.129	0.875097	0.515848	16.76921	0.967857	0.771565	20.02231	1.222334	0.611419
	+	15.86463			16.23019			24.47395		
*BLM*	−	9.167653	1.289576	0.494121	6.728003	1.382269	0.344308	6.201939	1.918412	0.039864
	+	11.82238			9.299909			11.89788		
*WRN*	−	7.16739	0.877522	0.533055	5.909568	0.799329	0.344161	5.718025	0.748446	0.237544
	+	6.289544			4.723689			4.279635		
*BRCA1*	−	17.60425	0.705496	0.162193	15.94288	1.06099	0.977051	15.8065	0.998662	0.843619
	+	12.41973			16.91523			15.78536		
*PARP1*	−	61.14282	0.895026	0.563022	47.50302	1.018003	0.908481	45.8812	0.970177	0.764114
	+	54.72441			48.35821			44.51287		
*E2F4*	−	11.49784	1.340332	0.414959	13.36074	0.680938	0.133485	10.49973	0.691647	0.14477
	+	15.41092			9.097842			7.262107		
*TP53*	−	16.52675	0.492513	0.009924	21.79782	0.203597	6.00 × 10^−8^	20.15492	0.292592	1.56 × 10^−5^
	+	8.139646			4.437968			5.897179		
*TERT*	−	0.647083	1.291257	0.635664	0.362969	1.783149	0.142159	0.187431	1.122226	1
	+	0.835551			0.647228			0.21034		

**Table 2 ijms-25-12151-t002:** Specific genes that were up-/down-regulated by CBCS treatment.

8 h	Up			Down		
	** *ATP1A3* **	** *ATP1B2* **	*ATP2A1*	*ATF1*	*DNMT1*	*DNMT3A*
	*ATP6V1C2*	*FOSB*	*HIST1H2AC*	*EP300*	*ELK4*	** *ETS2* **
	*HIST1H2AG*	*HIST1H2AG*	*HIST1H2AI*	** *LIG3* **	*MCM2*	*MCM3*
	*HIST1H2BD*	*HIST1H2BJ*	*HIST1H3J*	*MCM5*	*MCM7*	** *NFKB1* **
	*HIST1H4H*	*HIST2H2A4*	*HIST2H3C*	*PARP2*	*PARP3*	*PCNA*
	*HIST2H2BF*	*HIST2H3A*	*HIST2H3D*	*POLA1*	*POLE*	*POLE2*
				*POLF*	*RFC3*	** *TP53* **
				** *XBP1* **		
24 h	Up			Down		
	** *ATP1A3* **	** *ATP1B2* **	*CDKN1A*	*CDKN2A*	** *ETS2* **	*ERBB2*
	*FOS*	*GATA3*	** *IL1A* **	*LDHA*	*HK2*	** *NEIL1* **
	** *IL1B* **	** *IL6* **	** *IL8* **	** *NFKB1* **	*PARP3*	*POLE3*
	** *IL21R* **	** *JUN* **	** *NFATC1* **	*STAT5A*	*STAT5B*	*STAT6*
	*POLD4*	*STAT3*	*STAT4*	** *TP53* **	** *XBP1* **	
48 h	**Up**			**Down**		
	*BDNF*	*EGF*	*FGF18*	*APP3*	*LIG1*	** *LIG3* **
	** *IL1A* **	** *IL1B* **	** *IL6* **	** *NEIL1* **	** *NFKB1* **	** *TP53* **
	** *IL8* **	*IL12A*	*IL12B*			
	*IL18*	** *IL21R* **	*IL24*			
	** *JUN* **	** *NFATC1* **	*RELB*			

The transcripts that were evaluated as being significant at two time points or more are bold-typed.

**Table 3 ijms-25-12151-t003:** Primer pairs for amplifying the 5′-upstream regions of the *RB1* and *PARP1* genes.

Luc Plasmid	Primer	Sequence (5’ to 3’)
pGL4-RB1-Δ1	hRB1-3590	TTC**GGTACC**CACGCCAGGTTTCCCAG
	AhRB1-3787	AAT**CTCGAG**AAAAACCGGACGCGCCCTC
pGL4-RB1-Δ2	hRB1-3649	TTC**GGTACC**AGCGCCCCAGTTCCCCAC
	AhRB1-3787	AAT**CTCGAG**AAAAACCGGACGCGCCCTC
pGL4-RB1-Δ3	hRB1-3707	TTC**GGTACC**GCGGGCGGAAGTGACG
	AhRB1-3787	AAT**CTCGAG**AAAAACCGGACGCGCCCTC
pGL4-PARP1-Δ1	hPARP1-8202	TTC**GGTACC**CGGCAGGCGCCCGGGAAACTC
	AhPARP1-8041	AAT**CTCGAG**CCGCCACCGAACACGCCGC
pGL4-PARP1-Δ2	hPARP1-8135	TTC**GGTACC**GTGGACGCGGGTTCCGTGGGC
	AhPARP1-8041	AAT**CTCGAG**CCGCCACCGAACACGCCGC

*Kpn*I and *Xho*I recognition sequences are bold-typed.

## Data Availability

The RNAseq raw data for interpretation of the datasets will be made available only when permission is provided by the corresponding author, F.U. The RNAseq results comparing the experimental group (CBCS treatment) with the control after 8, 24, and 48 h are shown at the following links: https://tus.box.com/s/l0om5sh1t34tj6vfn22p2mxw25dmmff5 (accessed on 10 January 2025); https://tus.box.com/s/u0xgi6xv6sk7mouc0oi5wjviffy649sd (accessed on 10 January 2025); and https://tus.box.com/s/c1hxekion5lb2r62q5b553hjswj04tpo (accessed on 10 January 2025), respectively.

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
