# Peer review of "Effect of Culture Supernatant of Clostridium butyricum TO-A on Human DNA-Repair-Factor-Encoding Gene Promoters"

_ijms, 2024, doi:10.3390/ijms252212151_

Round 1
Reviewer 1 Report
Comments and Suggestions for Authors
In the manuscript named “Effect of culture supernatant of Clostridium butyricum TO-A on human DNA repair factor-encoding gene promoters”, Shunsuke Takaoka et al have cloned promoter regions of some genes, and the Luc reporters have showed they were regulated by CBCS. Meanwhile, authors have investigated their proteins and their expression with RNA-seq, these results would contribute to explore the molecular mechanism in prevention of carcinogenesis, which was useful for human healthy works in future. There were some comments about it.
(1) Authors have amplified approximately 500bp upstream of function genes, how, or why to select 500bp in this research.
(2) The Luc reporter results were not well confirmed by RNA-seq, and the RNA-seq results were also not confirmed by qRT-PCR, or other methods, please describe these results more clearly, and play more attention on Luc reporter expression and RNA-seq results.
(3) Statistical analysis section should be supplied with software information.
(4) Manuscript would be prepared with IJMS templet, with figures and tables in main text, not end of manuscript.
(5) In figure 7, the samples would not be clustered in present research, with order as “0-8-24-48” or “8-24-48” in two groups.
(6) The RNA-seq analysis needed to be described in detailly, for example, how to map reads to reference sequences, and how to quantify of gene expression.
(7) The “DO Enrichment”? the results were not shown, please check. The “Reactome database Enrichment” need refs.
(8) The conclusion was too long, please simply these sentences, for example, the line338 to line342 could be remove to discussion section, not here.
Author Response
Our responses to Comments and Suggestions
In the manuscript named “Effect of culture supernatant of Clostridium butyricum TO-A on human DNA repair factor-encoding gene promoters”, Shunsuke Takaoka et al have cloned promoter regions of some genes, and the Luc reporters have showed they were regulated by CBCS. Meanwhile, authors have investigated their proteins and their expression with RNA-seq, these results would contribute to explore the molecular mechanism in prevention of carcinogenesis, which was useful for human healthy works in future. There were some comments about it.
Thank you very much for reviewing our manuscript. We hope that it will contribute to progress in the field of medical sciences, especially the development of novel therapies or prevention of cancer.
(1) Authors have amplified approximately 500 bp upstream of function genes, how, or why to select 500 bp in this research.
We have been studying 5’-upstreams of human genes. Although not all, but lots of them have opposite direction transcribing gene pairs within 500-bp. These oppositely transcribed gene pairs are referred to as “bidirectional promoters”, implying that core-promoter functioning elements are present within 500-bp upstream of transcription start sites. Notably, numbers of the DNA-repair factor encoding gene promoters are bidirectional. We, therefore, just focused on the 500-bp regions. However, we cannot rule out the functions of enhancer elements, which might be distributed somewhere in the chromosomes, play roles in the formation of loop structures to regulate transcription.
(2) The Luc reporter results were not well confirmed by RNA-seq, and the RNA-seq results were also not confirmed by qRT-PCR, or other methods, please describe these results more clearly, and pay more attention on Luc reporter expression and RNA-seq results.
Thank you very much for the comment concerning the discrepancy between Luc assay (Fig. 1) and RNAseq analysis (Table 1). We have been using the human PIF1 promoter activity as a positive control standard for Luc assay. Because it gives relatively high Luc activity in HeLa cells but do not so much respond to trans-resveratrol and other drugs (https://novapublishers.com/shop/new-research-on-cell-aging-and-death/). In this study, Luc activities were normalized to that from pGL4-PIF1-transfected BCS-non-treated cells.
On the other hand, the RNAseq analysis, which contains a great majority of miRNA sequences, were not normalized to protein-encoding gene transcripts. If we normalize each FPKM value (Table 1) by that of PIF1I in both CBCS-treated and non-treated cells, we will find that the values of most of the gene transcripts would increase. Moreover, E2F4 and TP53 transcripts, which promoters show relatively low responses to CBCS treatment, were apparently decreased by CBCS (Table 1). However, the PARG and other gene transcripts, which promoters showed great positive responses to CBCS treatment (Fig. 1), were not so much reduced, suggesting that the up-regulated transcription initiation by promoter responses prevent from decreasing process. At present, the precise mechanism that causes an apparent decrease in the transcripts of some genes, including PIF1, is not known though. We presently speculate that it might come from the alterations in miRNA expression after n-butyric acid treatment.
Your comment is right and valuable suggesting that we should examine gene expression by qRT-PCR to confirm activation of DNA-repair associated gene promoters. However, through this study, we found that the effects from greatly up regulated miRNAs by n-butyric acid treatment on HeLa S3 cells. So that we would like to do the experiment sequentially with the analyses of key miRNAs that may regulate RNA metabolism.
(3) Statistical analysis section should be supplied with software information.
We have not used any special program, just carrying out the statistical analysis (T-TEST) on the raw data with the custom function of the excel software (Fig. 3).
(4) Manuscript would be prepared with IJMS template, with figures and tables in main text, not end of manuscript.
We have originally prepared manuscript on the IJMS template, though the formatting was done by editorial office. I guess the formatting was required for uploading it to “Preprints org” (https://www.preprints.org/manuscript/202410.0401/v1).
(5) In figure 7, the samples would not be clustered in present research, with order as “0-8-24-48” or “8-24-48” in two groups.
Thank you very much for the comment. Figure 7 is a heat map, which shows changes in expression of genes in CBCS-treated and non-treated HeLa S3 cells, made by Novogene Co. Ltd. However, in accordance with the comment we have edited Figure 7, arranging the order as 0-8-24-48 and 8-24-48.
(6) The RNA-seq analysis needed to be described in detail, for example, how to map reads to reference sequences, and how to quantify of gene expression.
Thank you very much for the comment. In this study, RNAseq analysis was carried out by Novogene Co. Ltd.
According to the analysis report, “Original image data file from high-throughput sequencing platforms (Illumina) was transformed to sequenced reads (Raw Data or Raw Reads) by CASAVA base recognition (Base Calling). Raw data were stored in FASTQ(fq) format files which contain sequences of reads and corresponding base quality. They were filtered to remove low quality reads or reads with adaptors.”
Regarding the quantification of transcripts, “Read counts is proportional to gene expression level. In this study, FPKM (Fragments Per Kilobase of transcript sequence per Millions base pairs sequenced) value was applied to estimate gene expression levels, which takes the effects into consideration of both sequencing depth and gene length on counting of fragments (Mortazavi et al., 2008).”
In the revised text, we just briefly described that RNAseq and the data analysis was done by Novogene, and FPKM value was used to estimate the gene expression adding the reference (Mortazavi et al., 2008).
(7) The “DO Enrichment”? the results were not shown, please check. The “Reactome database Enrichment” needs refs.
Thank you very much for the suggestion. In this study, we confirmed that “DO enrichment” was not done. Accordingly, it was deleted from the text. We added the reference [49] for the “Reactome”.
(8) The conclusion was too long, please simply these sentences, for example, the line338 to line342 could be remove to discussion section, not here.
We appreciate very much for the comment. We have edited the text, accordingly, reducing the conclusion section remaining a description for ncRNAs.
Reviewer 2 Report
Comments and Suggestions for Authors
This manuscript investigates whether CBCS and butyric acid affect the promoter activity of some DNA repair genes of HeLa S3 cells. Results suggest that the expression of some DNA repair genes is influenced by CBCS and butyric acid. This research provides useful information to study the biological role of these effect.
Major concerns:
1, Fig 3. Please perform the statistical assay to see whether the change is significantly different.
2, Fig5. As the band intensity of the loading control is also changing with the incubation time, it is difficult to draw the conclusion. The authors need to do a better experiment to obtain a very similar level of loading control.
3, RNA seq suggests that the expression of some genes is up-regulated and some are down-regulated. The authors need to perform experiments to confirm the result is valid with qRT-PCR or WB to confirm some key genes are regulated.
Comments on the Quality of English Language
English issues: It is very difficult to understand some sentences. such as: “In this study, it was revealed that C. butyricum TO-A culture supernatant (CBCS) increases several well-known DNA repair-related gene pro-51 moter activities in HeLa S3 cells the most.” change to "This study revealed that the culture supernatant of C. butyricum TO-A (CBCS) significantly enhances the promoter activities of several well-known DNA repair-related genes in HeLa S3 cells."
“In addition, n-butyric acid has been known to suppress the growth and inflammation of cancer cells, inducing cell morphology changes [13], cell death, and apoptosis of a colon cell line [14].” had better change to “ In addition, n-butyric acid is known to suppress cancer cell growth and inflammation, inducing changes in cell morphology, promoting cell death, and triggering apoptosis in a colon cell line.”
Other similar sentences need to be changed.
Author Response
Our responses to Comments and Suggestions
This manuscript investigates whether CBCS and butyric acid affect the promoter activity of some DNA repair genes of HeLa S3 cells. Results suggest that the expression of some DNA repair genes is influenced by CBCS and butyric acid. This research provides useful information to study the biological role of these effect.
Thank you very much for reviewing our manuscript. We hope that it will contribute to progress in the field of medical sciences, especially the development of novel therapies or prevention of cancer.
Major concerns:
1, Fig 3. Please perform the statistical assay to see whether the change is significantly different.
Thank you very much for the comment. Figure 3 was made from data of two independent experiments. We found another version including three independent experiments, with statistical analysis (N=3). Accordingly, the legend to Figure 3 was edited.
2, Fig 5. As the band intensity of the loading control is also changing with the incubation time, it is difficult to draw the conclusion. The authors need to do a better experiment to obtain a very similar level of loading control.
We appreciate very much for the comment. It was indicated that the signal of the b-actin is affected by the addition of CBCS (10%) to the culture medium. The amount of the b-actin in CBCS-treated cells increase at 1 to 8 h but declined after 32 h (Fig. 5). We agree with the comment though, it would be difficult because expression of great majority of the genes, including ncRNAs, are affected by CBCS (Fig. 7). Although quantification of each signal was not carried out, it would be able to conclude that ATM, RB1, and PARP1 proteins accumulate transiently after CBCS and n-butyrate addition more apparently that b-actin. However, presently, we are trying to find a protein that is not so much affected by the addition of n-butyric acid (5 mM) to the culture medium.
3, RNA seq suggests that the expression of some genes is up-regulated and some are down-regulated. The authors need to perform experiments to confirm the result is valid with qRT-PCR or WB to confirm some key genes are regulated.
Thank you very much for the comment suggesting that qRT-PCR analysis is needed to confirm the results of the RNAseq (Table 1).
RNAseq analysis, which does not only indicate expression of protein-encoding gene transcripts but also a great majority of miRNAs. It surely is, but if we select some gene transcript as a control, relative expression level after CBCS-treatment could be compared. As shown in Table 1, if we normalize each FPKM value by that of PIF1I in both CBCS-treated and non-treated cells, we will find that the values of most of the gene transcripts would increase. Moreover, E2F4 and TP53 transcripts, which promoters show relatively low responses to CBCS treatment, were apparently decreased by CBCS (Table 1). However, the PARG and other gene transcripts, which promoters showed great positive responses to CBCS treatment (Fig. 1), were not so much reduced, suggesting that the up-regulated transcription initiation by promoter responses prevent from decreasing process. At present, the precise mechanism that causes an apparent decrease in the transcripts of some genes, including PIF1, is not known though. We presently speculate that it might come from the alterations in miRNA expression after n-butyric acid treatment.
Your comment is right and valuable suggesting that we should examine gene expression by qRT-PCR to confirm activation of DNA-repair associated gene promoters. However, through this study, we found that the effects from greatly up regulated miRNAs by n-butyric acid treatment on HeLa S3 cells. So that we would like to do the experiment sequentially with the analyses of key miRNAs that may regulate RNA metabolism.
Regarding Quality of English Language
English issues: It is very difficult to understand some sentences. such as: “In this study, it was revealed that C. butyricum TO-A culture supernatant (CBCS) increases several well-known DNA repair-related gene promoter activities in HeLa S3 cells the most.” change to "This study revealed that the culture supernatant of C. butyricum TO-A (CBCS) significantly enhances the promoter activities of several well-known DNA repair-related genes in HeLa S3 cells."
“In addition, n-butyric acid has been known to suppress the growth and inflammation of cancer cells, inducing cell morphology changes [13], cell death, and apoptosis of a colon cell line [14].” had better change to “ In addition, n-butyric acid is known to suppress cancer cell growth and inflammation, inducing changes in cell morphology, promoting cell death, and triggering apoptosis in a colon cell line.”
Thank you very much for the comment. We have edited the text accordingly.
Other similar sentences need to be changed.
We appreciate very much for the comment. The original manuscript has been edited by a native English speaker, who lives in Oxford, UK. Surely, some sentences were further edited by co-authors. We, therefore, concentrated on checking such modifications, especially “Conclusion” section.
Round 2
Reviewer 2 Report
Comments and Suggestions for Authors
The authors answered my concerns. I have no concern for this manuscript.